# PTX3 Modulates Neovascularization and Immune Inflammatory Infiltrate in a Murine Model of Fibrosarcoma

**DOI:** 10.3390/ijms20184599

**Published:** 2019-09-17

**Authors:** Tiziana Annese, Roberto Ronca, Roberto Tamma, Arianna Giacomini, Simona Ruggieri, Elisabetta Grillo, Marco Presta, Domenico Ribatti

**Affiliations:** 1Department of Basic Medical Sciences, Neurosciences and Sensory Organs, Section of Human Anatomy and Histology, University of Bari Medical School Bari, P.zza Giulio Cesare 11, 70124 Bari, Italy; tiziana.annese@uniba.it (T.A.); roberto.tamma@uniba.it (R.T.); simona.ruggieri@uniba.it (S.R.); 2Department of Molecular and Translational Medicine, University of Brescia, V.le Europa, 11, 25123 Brescia, Italy; roberto.ronca@unibs.it (R.R.); arianna.giacomini@unibs.it (A.G.); elisabetta.grillo@unibs.it (E.G.)

**Keywords:** angiogenesis, FGF-2, fibrosarcoma, inflammation, PTX-3

## Abstract

Fibrosarcoma is an aggressive subtype of soft tissue sarcoma categorized in infantile/congenital-type and adult-type. Fibrosarcoma cells and its surrounding immune inflammatory infiltrates overexpress or induce the expression of fibroblast growth factor-2 (FGF-2) that have a crucial role in tumor progression and angiogenesis. The inflammation-associated long pentraxin 3 (PTX3) was found to reduce FGF-2-mediated angiogenesis, but its role on fibrosarcoma immune inflammatory infiltrate is still unknown. In this study, we have evaluated the PTX3 activity on immune infiltrating mast cells, macrophages and T-lymphocytes by immunohistochemistry on murine MC-TGS17-51 fibrosarcoma cells and on transgenic TgN(Tie2-hPTX3) mouse. In these fibrosarcoma models we found a reduced neovascularization and a significant decrease of inflammatory infiltrate. Indeed, we show that PTX3 reduces the level of complement 3 (C3) deposition reducing fibrosarcoma progression. In conclusion, we hypothesize that targeting fibrosarcoma microenvironment by FGF/FGFR inhibitors may improve treatment outcome.

## 1. Introduction

Fibrosarcoma originates from transformed spindle-shaped fibroblasts [1]. It occurs as a soft-tissue mass or as a primary or secondary bone tumor, predominantly located in deep soft tissues or adjacent to bones, even though it may occur at any anatomic site. Fibrosarcoma can be divided into infantile/congenital fibrosarcoma, with intermediate malignant and rarely metastasizing features, and adult-type fibrosarcoma, with more aggressive and malignant features [2].

In a transgenic mouse model, dermal fibrosarcomas develop in three stages: mild fibromatosis, aggressive fibromatosis, and fibrosarcoma, with the latter two stages being highly vascularized when compared with both the normal dermis and the initial mild lesion [3]. Analysis of fibroblasts derived from biopsies of these lesions revealed that fibroblast growth factor-2 (FGF-2) is synthesized at all three stages and by normal dermal fibroblasts, FGF-2 expression being correlated to angiogenesis and tumor progression [3].

Long pentraxin-3 (PTX3) is a member of the pentraxin family produced locally in response to inflammatory signals [4]. PTX3 was found to bind FGF-2 and inhibiting FGF-2-dependent endothelial cell proliferation in vitro and angiogenesis in vivo [5,6,7]. Transgenic PTX3 overexpression by tumor cells impairs the activation of the FGF/FGF receptor (FGFR) system in FGF-driven tumor cell lines, affecting tumor growth and metastasis in different models of melanoma, prostate, and mammary carcinomas [8,9,10]. PTX3 accumulation in tumor stroma and bloodstream obtained through endothelial specific overexpression of PTX3 in transgenic mice, affects tumorigenic, angiogenic, and metastatic potential of various syngeneic FGF-dependent tumor cell lines [11,12].

Recently, Rodrigues et al. [13] demonstrated that PTX3 overexpression significantly reduced the proliferative and tumorigenic potential of fibrosarcoma cells in vitro and in vivo. Moreover, treatment with the PTX3-derived FGF-trap small molecule NSC12 caused a significant inhibition of the tumorigenic potential of fibrosarcoma cells, paralleled by a decrease of FGFR activation and signaling in these cells. In addition, we demonstrated that PTX3 overexpression in transgenic mice or treatment with the FGF inhibitor NSC12 result in a significant inhibition of the growth and vascularization of TRAMP-C2 tumor grafts, a murine model of prostate cancer, that were paralleled by a decrease of mast cell infiltrate into the lesion [14].

There is clear evidence that different components of the innate and adaptive immunity, including lymphocytes, mast cells, and macrophages, play an active and coordinate role in enhancing tumor angiogenesis [15]. In this study, we have evaluated and characterized the immune infiltrating cell content of murine fibrosarcoma grafts represented by T-lymphocytes, mast cells, and macrophages, and correlated it with the vascular density of the lesion following tumor cell- or transgenic/host-mediated PTX3 overexpression.

## 2. Materials and Methods

### 2.1. Cell Culture

Murine MC-TGS17-51 (MC17-51) fibrosarcoma cells were obtained from American Type Culture Collection (ATCC) and were cultured at 37 °C with 5% CO_2_ in DMEM (Gibco) containing penicillin/streptomycin (100 U and 10 mg/mL, respectively) and supplemented with 10% fetal bovine serum (FBS-Gibco). Fibrosarcoma cells were transfected with a pBABE-Puro vector harboring the full length human PTX3 cDNA (GenBank accession n◦ X63613) or a pBABE-Puro empty vector (mock) using FuGENE (Promega). MC17-51 stable transfectants were selected in the presence of 1.0 µg/mL of puromycin.

### 2.2. In Vivo Procedures

A total of 14 seven-week-old C57BL/6 male mice were injected subcutaneously (s.c.) into the dorsolateral flank with mock- or PTX3-overexpressing murine MC17-51 cells (1 × 10^6^). In an additional experiment, wild-type C57BL/6 and transgenic TgN(Tie2-hPTX3) mice were injected s.c. with wild type MC17-51 cells (1 × 10^6^). Tumor growth was followed and grafts were removed at the end of the experimental procedure, weighted and paraffin embedded for immunohistochemistry (IHC). All animal experiments were approved by the local animal ethics committee (OPBA, Organismo Preposto al Benessere degli Animali, Università degli Studi di Brescia, Brescia, Italy) and were performed in accordance with national guidelines and regulations. Procedures involving animals and their care conformed to institutional guidelines that comply with national and international laws and policies (EEC Council Directive 86/609, OJ L 358, 12 December 1987) and with “ARRIVE” guidelines (Animals in Research Reporting in vivo Experiments).

### 2.3. Immunohistochemistry

Three-micrometer-thick, 4% paraformaldehyde (PFA) fixed and paraffin-embedded histological sections collected on poly-L-lysine-coated slides (code J2800AMNZ, Gerhard Menzel, Germany) were deparaffinized and rehydrated in a xylene-graded alcohol scale and then rinsed for 10 min in Tris buffered saline solution (TBS, code SRE0032, Sigma-Aldrich, St. Louis, MO, USA). For CD31 and CD68 staining the following protocol was performed: the sections were heated in a solution of sodium citrate pH 6.0 (code S1699, Agilent Dako, Santa Clara, CA USA) once the temperature reached 96 °C in a water bath for 20 min; after washing in TBS + 0.025 Triton X-100 (TBS-T), the slides were kept 2 h in a blocking buffer [BB; TBS pH 7.4 + 1% bovine serum albumin (BSA) + 10% normal goat serum (NGS)]; afterwards, the sections were exposed to rabbit polyclonal-CD31 (diluted 1:60; code ab28364, Abcam, Cambridge, UK) or rabbit polyclonal-CD68 (diluted 1:100; code ab125047, Abcam, Cambridge, UK) primary unconjugated antibodies diluted in TBS + 1% BSA overnight at 4°C; after washings in TBS-T, the endogenous peroxidase was blocked in 3% hydrogen peroxide for 10 min in the dark; then, the sections were incubated 1 h with biotinylated goat anti-rabbit IgG (code BA-1000, Vector Laboratories, Burlingame, CA, USA) diluted 1:150 in TBS + 1% BSA follow by streptavidin-peroxidase conjugate (code A-2704, Vector Laboratories, Burlingame, CA, USA) for 30 min; immunodetection was performed in distillate water with an AEC substrate kit for peroxidase (code SK-4200, Vector Laboratories, Burlingame, CA, USA) for 40 min at room temperature.

For Tryptase staining the following protocol was performed: the sections were heated in a solution of sodium citrate pH 6.0 (code S1699, Agilent Dako, Santa Clara, CA, USA) at 98 °C in a water bath for 30 min; after washing in TBS-T, the endogenous peroxidase was blocked in 3% hydrogen peroxide for 15 min in the dark; the sections were kept 30 min in a rodent block (from Mouse on Mouse Polymer IHC Kit, code ab127055, Abcam, Cambridge, UK). Afterwards, the sections were exposed to mouse monoclonal-Tryptase (diluted 1:250; code ab2378, Abcam, Cambridge, UK) primary unconjugated antibody diluted in TBS + 1% BSA overnight at 4°C. After washings in TBS-T, the sections were incubated 15 min with Mouse on Mouse Polymer IHC in the dark (code ab127055, Abcam, Cambridge, UK). The immunodetection was performed in distillate water with AEC substrate kit for peroxidase (code SK-4200, Vector Laboratories, Burlingame, CA, USA) for 40 min at room temperature.

For C3, PTX3, CD4, and CD8 staining, the following protocol was performed: the sections were heated in a solution of Tris/EDTA pH 9.0 (code S2367, Agilent Dako, Santa Clara, CA, USA) and in a solution of sodium citrate pH 6.0 (code S1699, Agilent Dako, Santa Clara, CA, USA), for C3/CD4 and for PTX3/CD8, respectively, once the temperature reached 98 °C in a water bath for 30 min. After washing in TBS-T, the endogenous peroxidase and alkaline phosphatase were blocked in Dual Endogenous Enzyme-Blocking target (code S2003, Agilent Dako, Santa Clara, CA, USA) for 10 min; afterwards, the sections were exposed to rabbit monoclonal-C3 (diluted 1:2000; code ab200999, Abcam, Cambridge, UK) rabbit polyclonal-PTX3 (diluted 1:500; (kind gift of Prof. Mantovani, Humanitas Clinical Institute, Milan, Italy), rabbit monoclonal-CD4 (diluted 1:750; code ab183685, Abcam, Cambridge, UK), or rabbit polyclonal-CD8 (diluted 1:250; code ab203035, Abcam, Cambridge, UK) primary unconjugated antibodies diluted in anybody diluent (code ab64211, Abcam, Cambridge, UK) for 30 min at room temperature. After washings in TBS-T, the sections were incubated with Dako REAL™ Detection System, Alkaline Phosphatase/RED, Link, Biotinylated Secondary Antibodies (AB2) for 15 min (code K5005, Agilent Dako, Santa Clara, CA, USA); then, with Dako REAL™ Detection System, Alkaline Phosphatase/RED Streptavidin Alkaline Phosphatase (AP) (code K5005, Agilent Dako, Santa Clara, CA, USA) for 15 min. Finally, the immunodetection was performed with Dako REAL™ Detection System, Chromogen (RED) (code K5005, Agilent Dako, Santa Clara, CA, USA) for 20 min at room temperature.

At the end of secondary antibodies revelation, all the sections were washed in distillate water, counterstained with Mayer’s hematoxylin (code 51275, Sigma-Aldrich, St. Louis, MO, USA), and mounted in glycergel (code C0563, Agilent Dako, Santa Clara, CA, USA). Specific pre-immune serum replacing the primaries antibodies served as a negative control.

### 2.4. Morphometric Analysis

Morphometric analysis was performed by two independent observers on three randomly selected fields from seven animals per subgroup. Immunohistochemistry slides were scanned using the whole-slide scanning platform Aperio ScanScope CS (Leica Biosystems, Nussloch, GmbH Germany) at the maximum magnification available (40×), stored as digital high resolution images on the workstation associated with the instrument, and analyzed using the Aperio Positive Pixel Count algorithm embedded in the ImageScope v.11.2.0.780 (Leica Biosystems, Nussloch, GmbH Germany).

### 2.5. Statistical Analysis

Results are given as mean ± SD. Statistical analysis were performed using two-way ANOVA and Bonferroni post-hoc tests to compare replicate with GraphPad Prism 5.01 statistic package (GraphPad Software, San Diego, CA, USA) and *p* < 0.05 was considered as the limit for statistical significance.

## 3. Results

PTX3 may exert a significant impact on tumor growth and angiogenesis in different tumor types, and has been reported to play a relevant role in the regulation and recruitment of innate immune cells [11]. However, no data are available on the possible correlation among PTX3 expression, tumor growth, angiogenesis, and immune infiltrate in regulating soft tissue sarcomas.

In order to evaluate the effect of PTX3 expression on fibrosarcoma growth and to characterize its neovascular response and inflammatory infiltrate profile, we took advantage of a murine syngeneic fibrosarcoma cell line (MC17-51) (American Type Culture Collection [ATCC] clone CRL-2799; ATCC, Manassas, VA, USA) and of a transgenic TgN (Tie2-hPTX3) mouse model characterized by the endothelial-specific expression of PTX3 driven by the mouse *endothelial-specific receptor tyrosine kinase* (Tie2) promoter the Tie promoter [12]. In these mice, the production of PTX3 by endothelial cells leads to the accumulation of the protein in the blood stream and stroma of all the organs examined with no apparent signs of toxicity. Thus, this model allows investigating the impact of systemic expression of PTX3 protein in vivo along the different phases of tumor take and progression and its role in tumor-stroma cross talk in FGF-dependent tumors [12].

As already reported [13], murine fibrosarcoma MC17-51 cells, that express very low levels of PTX3, were transfected with a pBABE-Puro vector, possibly harboring the full length human PTX3 cDNA sequence, to generate PTX3-overexpressing MC17-51 (PTX3-MC17-51) or control/mock (mock-MC17-51) cells, respectively.

To evaluate the effects of PTX3 expression on tumor growth and to characterize angiogenesis and the inflammatory infiltrate, mock- and PTX3-overexpressing MC17-51 cells were injected s.c. in the flank of C57BL/6 mice. Likewise, wild type MC17-51 cells were grafted in wild type (WT) and transgenic TgN (Tie2-hPTX3) mice.

As shown in Figure 1A, the overexpression of PTX3 by PTX3-MC17-51 cells caused a significant reduction of tumor growth when compared to wild type MC17-51 grafts, as demonstrated by the reduced tumor weight measured at the end of the experimental procedure. Similar results were obtained when wild type MC17-51 cells were grafted in transgenic TgN(Tie2-hPTX3) mice and compared to wild type MC17-51 lesions growing in WT animals (Figure 1B). IHC on tumor specimens confirmed a strong positivity for PTX3 in PTX3-MC17-51 samples (Figure 1D) and in MC17-51 tumors grown in TgN(Tie2-hPTX3) mice (Figure 1G) when compared to the corresponding controls (Figure 1C,F).

Next, all fibrosarcoma samples obtained following grafting of PTX3-MC17-51 or mock-MC17-51 cells in syngeneic mice or from wild type MC17-51 tumors generated in WT and transgenic TgN(Tie2-hPTX3) mice were evaluated for their neovascular response and immune inflammatory infiltrate by IHC.

As shown in Figure 2, PTX3 overexpression caused a significant reduction of tumor angiogenesis/CD31^+^ areas. This was observed both when PTX3-MC17-51 grafts in syngeneic animals were compared to mock-MC17-51 lesions (Figure 2A–C) and when wild type MC17-51 tumors growing in TgN(Tie2-hPTX3) animals were compared to the lesions occurring in WT mice (Figure 2D–F). Morphometric data are reported in Figure 2C,F and in Table 1.

In parallel with the inhibition of tumor-associated neovascularization, PTX3 overexpression resulted in a significant reduction in the recruitment of immune cell populations such as mast cells, macrophages, and T-lymphocytes (Figure 3, Figure 4, Figure 5 and Figure 6). Morphometric data are reported in Figure 3, Figure 4, Figure 5 and Figure 6C,F and in Table 1.

Overall, these data strongly correlate the tumor-impairing activity of PTX3 expressed by tumor cells or by tumor stroma with a reduced neovascular response and a significant decrease of inflammatory infiltrate in these fibrosarcoma models.

The complement component 3 (C3) plays a central role in the complement system and contributes to innate immunity. PTX3 knockout goes along with increased C3 deposition in neoplastic tissues that correlates with an increased tumor progression [16]. On this basis, C3 deposition was quantified in all fibrosarcoma samples. As shown in Figure 7, PTX3 overexpression resulted in a significant reduction in the levels of C3 immunoreactivity in fibrosarcoma grafts. Indeed, PTX3-overexpressing PTX3-MC17-51 tumors (Figure 7B) and wild type MC17-51 tumors grafted in TgN(Tie2-hPTX3) mice (Figure 7E) are characterized by low amounts of C3 protein when compared to mock-MC17-51 lesions (Figure 7A) or to wild type MC17-51 tumors grafted in WT animals (Figure 7D). Morphometric data are reported in Figure 7C,F and in Table 1.

Together, these data show that PTX3 overexpression exerts a relevant anti-tumorigenic activity in fibrosarcoma accompanied by a significant conditioning of tumor microenvironment in terms of angiogenesis and inflammatory infiltrate.

## 4. Discussion

Literature evidences have shown a close relationship between angiogenesis and cancer initiation and progression. Indeed, highly vascularized tumors have an increased risk of metastasis and a poorer prognosis [17]. The tumor microenvironment contains a heterogeneous and complex mixture of stromal cells including fibroblasts, pericytes, endothelial, mesenchymal, and cells of the hematopoietic system including lymphocytes, mast cells, and macrophages that actively support tumor growth and angiogenesis [17,18].

To date, little literature evidence concerns itself with the relationship between angiogenesis, inflammatory infiltrate, and tumor progression in fibrosarcoma. The proangiogenic, pleiotropic FGF-2 is synthesized by normal dermal fibroblasts and at all the stages of development of a transgenic mouse model of fibrosarcoma, its expression being correlated to angiogenesis and tumor progression [3]. A close relationship has been established between tumor progression and mast cell infiltrate in an experimental model of fibrosarcoma [19]. Eisenthal et al. [20] demonstrated an anti-angiogenic activity of TNF-α associated to a reduced tumor growth in a murine experimental model of fibrosarcoma. In a retrospective study, Jones et al. [21] showed that anti-angiogenic therapy provided prolonged clinical benefit in patients with advanced/metastatic chondrosarcoma.

Interestingly, PTX3 expression has been described as epigenetically downregulated in various types of tumors, including fibrosarcoma, and its overexpression has been proven to be sufficient to reduce tumor burden [22]. Even though it is challenging to define the mechanism(s) by which PTX3 may act as an oncosuppressor, its FGF-trap activity and its capacity to modulate the “immune signature” have been reported [22].

In this study, we have evaluated the effect of PTX3 on tumor vascular density and mast cell, macrophage and T-lymphocyte infiltrates in a murine fibrosarcoma model in which PTX3 was overexpressed by grafted fibrosarcoma cells or by the endothelium of the host animals.

In agreement with the anti-tumor activity exerted by both tumor cell- or stroma-derived PTX3, tumor neovascularization was significantly reduced in the presence of PTX3. Because of PTX3 overexpression, tumor samples were also characterized by a significant reduction of CD4^+^ and CD8^+^ lymphocyte, mast cell and CD68^+^ macrophage pro-inflammatory populations.

Genetic ablation of PTX3 results in a factor H-mediated increase of C3 complement component deposition, thus leading to a pro-tumor inflammatory context that confers increased susceptibility to mesenchymal and epithelial carcinogenesis [16]. Here, we found that the reduced inflammatory infiltrate observed in PTX3 overexpressing tumors was paralleled by a strong reduction of C3 deposition in the fibrosarcoma specimens analyzed, further confirming the role of PTX3 in the modulation of tumor-associated inflammation that, in turn, may affect tumor neovascularization and growth. Notably, the impact of PTX3 on fibrosarcoma microenvironment results in a decrease of immune/inflammatory components regardless from its cellular source (i.e., grafted tumor cells or endothelium of the host). Indeed, PTX3-MC17-51 grafts in WT mice and wild type MC17-51 lesions growing in TgN (Tie2-hPTX3) mice showed a similar profile of infiltrating cells.

Therapeutic strategies aimed to reduce the recruitment or to “orientate” immune cells in the tumor microenvironment are of potential therapeutic interest because of their impact on tumor growth, immune-tolerance and angiogenesis. In this frame, the study of PTX3 activity in the tumor milieu may represent a prototype of how a deeper analysis of tumor microenvironment, following treatments or modulation of stromal components, may unveil significant effects or add more details on the evolution of tumor populating cells. Selective ablation of immune cell populations or complement components will provide more detailed information about their individual contribution to reduce tumor vascularization and/or growth in fibrosarcoma. Finally, notwithstanding the complexity of the mechanism of action of PTX3 on different molecular and cellular components of the tumor and host, our results support the hypothesis that anti-FGF/FGFR strategies may exert a therapeutic effect in fibrosarcoma.

## Figures and Tables

**Figure 1 ijms-20-04599-f001:**
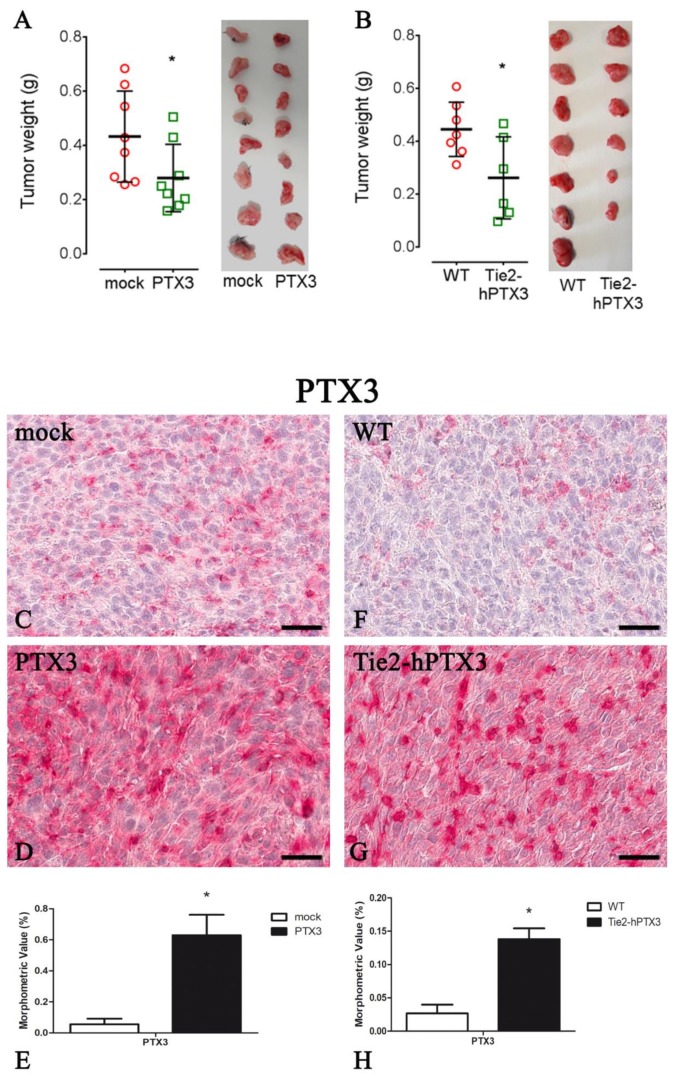
Pentraxin-3 (PTX3_ overexpression reduces tumor growth. Tumors weight (**A**,**B** left panel) and representative tumors images (A,B right panel) at the end of the experiment in mock (**A**) and PTX3 (**A**) transfected MC17-51 injected subcutaneously (s.c.) in syngeneic C57BL/6 mice and in WT MC17-51 cells injected s.c. in wild type (C57BL/6) (**B**) and transgenic TgN(Tie2-hPTX3) (**B**) mice. PTX3 immunohistochemistry and morphometric analysis in mock (**C**) and PTX3 (**D**) transfected MC17-51 injected s.c. in syngeneic C57BL/6 mice and in WT MC17-51 cells injected s.c. in wild type (C57BL/6) (F) and transgenic TgN(Tie2-hPTX3) (**G**) mice. Morphometric analysis shows a significant decrease of PTX3 content in PTX3 (E) and TgN(Tie2-hPTX3) (**H**) compared to their respective controls. * *p* < 0.05. Scale bar: C, D, F, G 60 µm.

**Figure 2 ijms-20-04599-f002:**
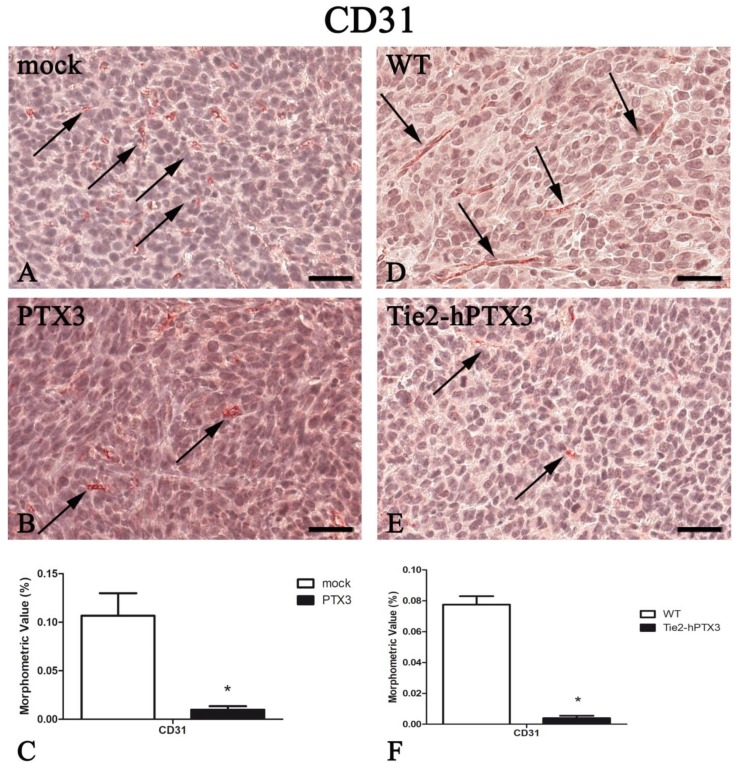
PTX3 overexpression reduces angiogenesis. CD31 immunohistochemistry and morphometric analysis in mock (**A**) and PTX3 (**B**) transfected MC17-51 tumor grafts growing s.c. in syngeneic C57BL/6 mice and in wild type MC17-51 tumor grafts growing s.c. in WT C57BL/6 (**D**) and transgenic TgN(Tie2-hPTX3) mice (**E**) mice. Morphometric analysis shows a significant decrease of CD31 positive vessels (arrows) following PTX3 overexpression by tumor cells (**C**) or by transgenic tumor stroma (**F**) when compared to their corresponding controls (**A**,**D**, respectively). * *p* < 0.05. Scale bar: **A**,**B**,**D**,**E** 60 µm.

**Figure 3 ijms-20-04599-f003:**
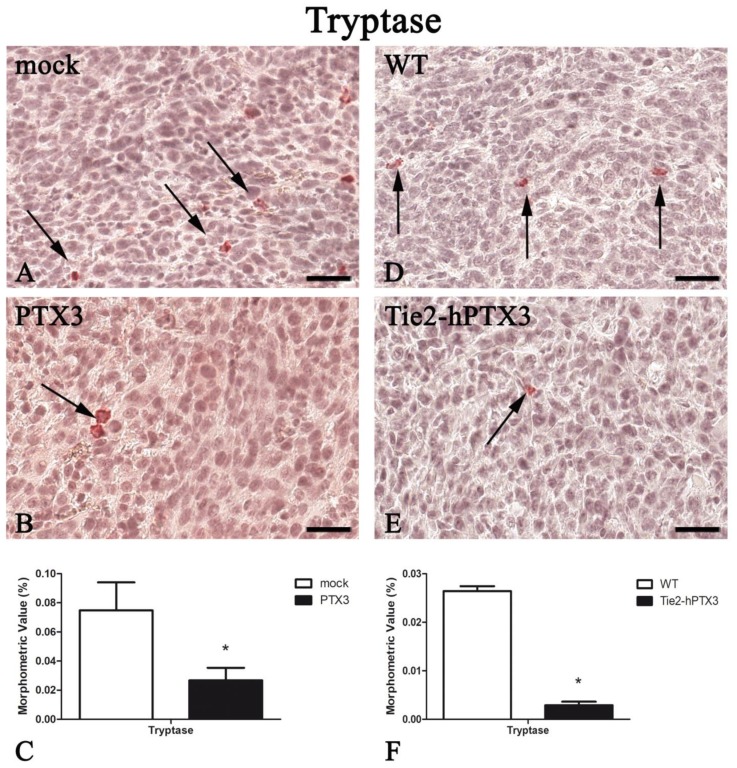
PTX3 overexpression reduces Tryptase-positive mast cells infiltrate. Tryptase immunohistochemistry and morphometric analysis in mock (**A**) and PTX3 (**B**) transfected MC17-51 tumor grafts growing s.c. in syngeneic C57BL/6 mice and in wild type MC17-51 tumor grafts growing s.c. in WT C57BL/6 (**D**) and transgenic TgN(Tie2-hPTX3) mice (**E**) mice. Morphometric analysis shows a significant decrease of Tryptase-positive mast cells (arrows) following PTX3 overexpression by tumor cells (**C**) or by transgenic tumor stroma (**F**) when compared to their corresponding controls (**A**,**D**, respectively). * *p* < 0.05. Scale bar: (**A**,**B**,**D**,**E**) 60 µm.

**Figure 4 ijms-20-04599-f004:**
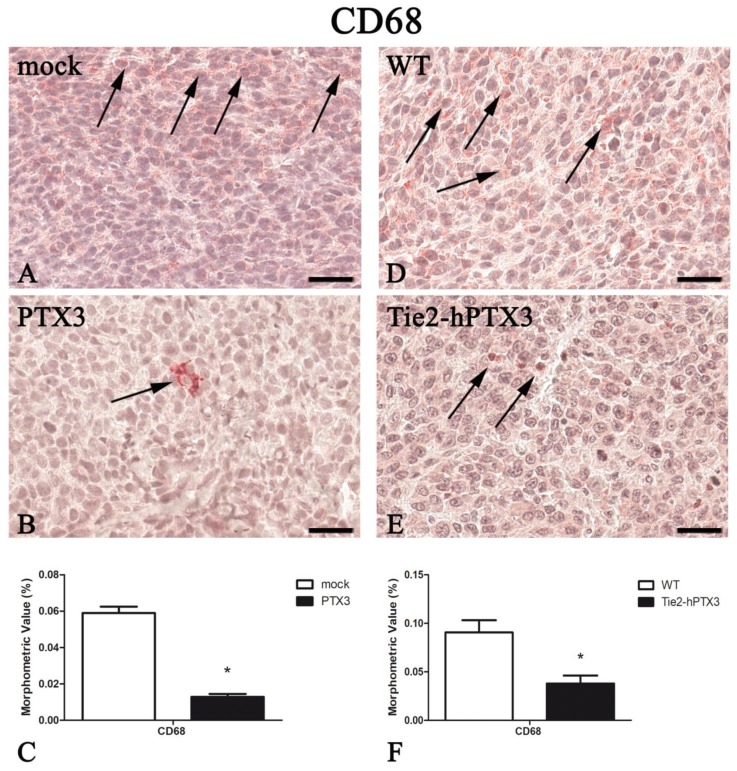
PTX3 overexpression reduces CD68-positive macrophages infiltrate. CD68 immunohistochemistry and morphometric analysis in mock (**A**) and PTX3 (**B**) transfected MC17-51 tumor grafts growing s.c. in syngeneic C57BL/6 mice and in wild type MC17-51 tumor grafts growing s.c. in WT C57BL/6 (**D**) and transgenic TgN(Tie2-hPTX3) mice (**E**) mice. Morphometric analysis shows a significant decrease of CD68-positive macrophages (arrows) following PTX3 overexpression by tumor cells (**C**) or by transgenic tumor stroma (**F**) when compared to their corresponding controls (**A**,**D**, respectively). * *p* < 0.05. Scale bar: (**A**,**B**,**D**,**E**) 60 µm.

**Figure 5 ijms-20-04599-f005:**
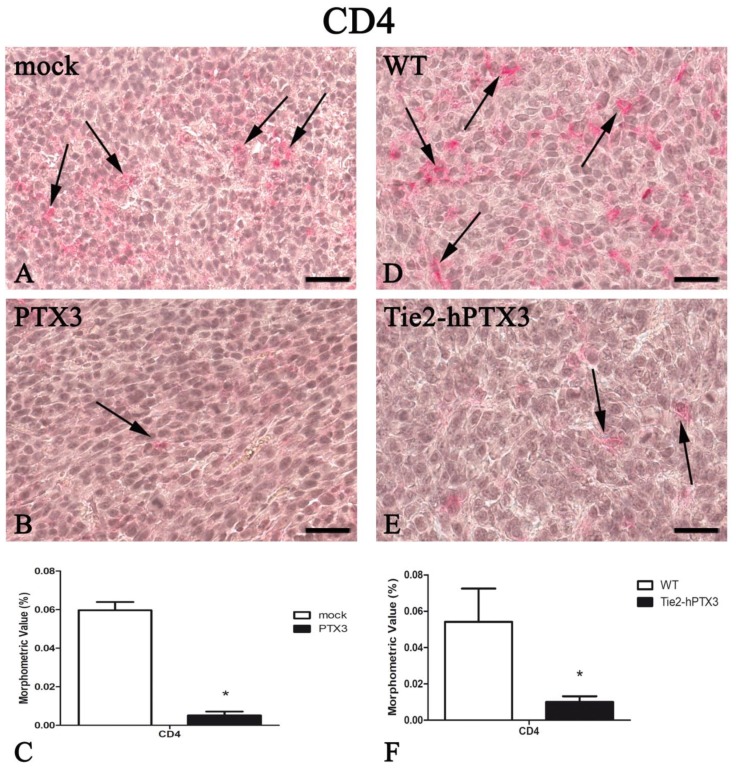
PTX3 overexpression reduces CD4-positive T-lymphocytes infiltrate. CD4 immunohistochemistry and morphometric analysis in mock (**A**) and PTX3 (**B**) transfected MC17-51 injected s.c. in syngeneic C57BL/6 mice and in WT MC17-51 cells injected s.c. in wild type (C57BL/6) (**D**) and transgenic TgN(Tie2-hPTX3) (**E**) mice. Morphometric analysis shows a significant decrease of CD4-positive T-lymphocytes (arrows) following PTX3 overexpression by tumor cells (**C**) or by transgenic tumor stroma (**F**) when compared to their corresponding controls (**A**,**D**, respectively). * *p* < 0.05. Scale bar: (**A**,**B**,**D**,**E**) 60 µm.

**Figure 6 ijms-20-04599-f006:**
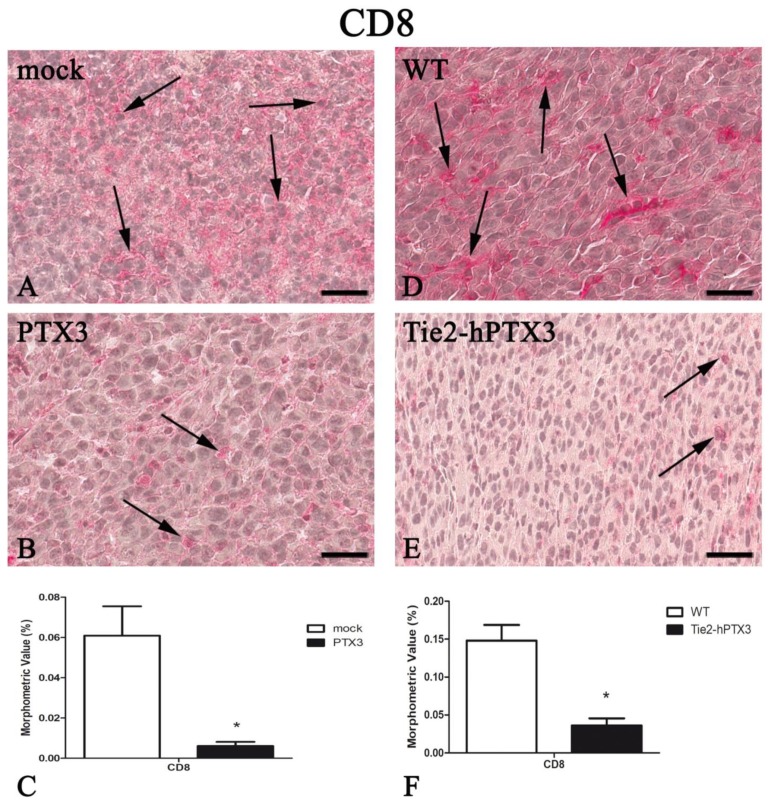
PTX3 overexpression reduces CD8-positive T-lymphocytes infiltrate. CD8 immunohistochemistry and morphometric analysis in mock (**A**) and PTX3 (**B**) transfected MC17-51 injected s.c. in syngeneic C57BL/6 mice and in WT MC17-51 cells injected s.c. in wild type (C57BL/6) (**D**) and transgenic TgN(Tie2-hPTX3) (**E**) mice. Morphometric analysis shows a significant decrease of CD8-positive T-lymphocytes (arrows) following PTX3 overexpression by tumor cells (**C**) or by transgenic tumor stroma (**F**) when compared to their corresponding controls (**A**,**D**, respectively). * *p* < 0.05. Scale bar: (**A**,**B**,**D**,**E**) 60 µm.

**Figure 7 ijms-20-04599-f007:**
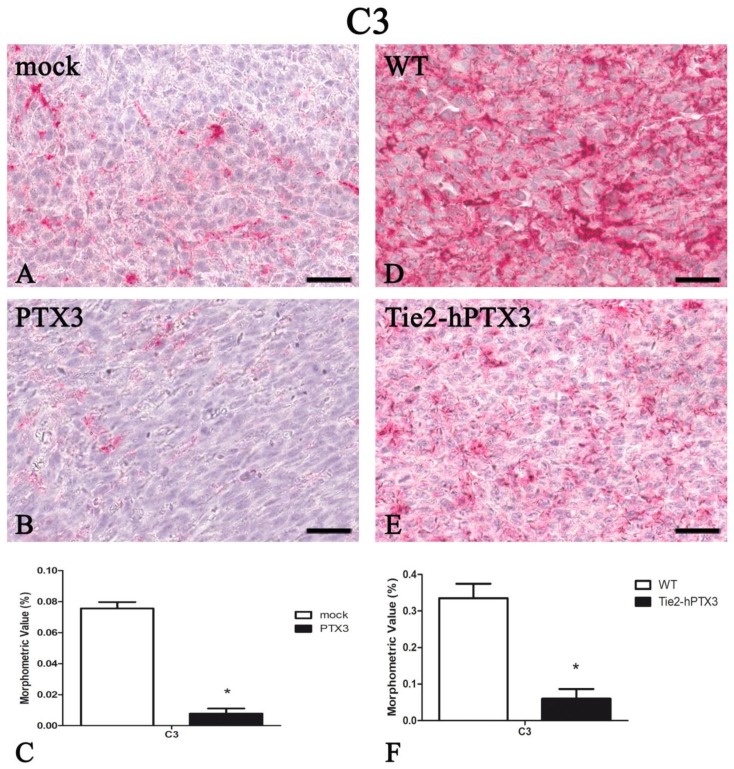
PTX3 overexpression reduces the deposition of the complement C3 component. C3 immunohistochemistry and morphometric analysis in mock (**A**) and PTX3 (**B**) transfected MC17-51 injected s.c. in syngeneic C57BL/6 mice and in WT MC17-51 cells injected s.c. in wild type (C57BL/6) (**D**) and transgenic TgN(Tie2-hPTX3) (**E**) mice. Morphometric analysis shows a significant decrease of C3 tumor content following PTX3 overexpression by tumor cells (**C**) or by transgenic tumor stroma (**F**) when compared to their corresponding controls (**A**,**D**, respectively). * *p* < 0.05. Scale bar: (**A**,**B**,**D**,**E**) 60 µm.

**Table 1 ijms-20-04599-t001:** Summary of the morphometric analysis performed on immunohistochemistry slides.

Protein	PTX3 Overexpression by Cancer Cells		PTX3 Overexpression by Transgenic Animals	
	mock	PTX3	WT	Tie2-hPTX3
PTX3	0.055 ± 0.036	0.630 ± 0.131 *	0.027 ± 0.013	0.138 ± 0.016 *
CD31	0.107 ± 0.023	0.010 ± 0.004 *	0.077 ± 0.005	0.004 ± 0.002*
Tryptase	0.075 ± 0.019	0.027 ± 0.009 *	0.026 ± 0.001	0.003 ± 0.001*
CD68	0.059 ± 0.004	0.013 ± 0.002 *	0.091 ± 0.013	0.038 ± 0.008*
CD4	0.060 ± 0.004	0.005 ± 0.002 *	0.054 ± 0.018	0.010 ± 0.003*
CD8	0.061 ± 0.015	0.006 ± 0.002 *	0.148 ± 0.021	0.036 ± 0.010*
C3	0.076 ± 0.004	0.008 ± 0.003 *	0.335 ± 0.039	0.060 ± 0.027*

Data of the morphometric values (%) obtained for the indicated antigenes are expressed as mean ± SD (*n* = 7) and have been utilized for the figures of this work. Bonferroni post-test was used to compare all groups after a two-way ANOVA (* *p* < 0.001).

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
