# Peer review of "PTX3 Modulates Neovascularization and Immune Inflammatory Infiltrate in a Murine Model of Fibrosarcoma"

_ijms, 2019, doi:10.3390/ijms20184599_

Round 1
Reviewer 1 Report
The authors present a thorough mechanistic investigation on the pathophysiological role of PTX3 in the carcinogenesis, immunization and angiogenesis of fibrosarcoma. The research is mostly done in vivo and thoroughly designed and reported. As such, the study is certainly of interest for the readers of this journal. Some minor issues should be improved before publication is warranted:
1. The abstract has a minor typo: "The inflammation-associated long 27 pentraxin 3 (PTX3) was found to reduceS FGF-2-mediated angiogenesis"
2. Several sentences in the results section are borderline unreadable because the authors have chosen to include many means and standard deviations in the results text, e.g. "Indeed, the mean±SD of infiltrating Tryptase-positive mast cell morphometric values (Fig. 3) 137 were: 0.075±0.019 vs 0.027±0.009 in mock- vs PTX3-MC17-51 grafts (Fig. 3C) and 0.026±0.001 vs 138 0.003±0.001 for wild type MC17-51 lesions growing in WT vs TgN(Tie2-hPTX3) mice (Fig. 3F)." My suggestion is that it would be better to include these numbers as a small table in the figures itself so that the results text can be kept as clean as possible.
3. In the introduction or results section, the authors should put more effort in describing the experimental relevance of the TgN(Tie2-hPTX3) mouse model. In the current version, this model is not explained and therefore readers may not fully understand the relevance of these experiments.
4. In the title of the last figure, the authors write "Figure 7. PTX3 overexpression reduces apoptosis.". However, theyhey do not explain what the link between C3 and apoptosis would be. In addition, it is unclear to me how reduced apoptosis would lead to reduced tumor progression.
5. The authors could more explicitly describe in the discussion that their results are largely correlative and what additonal experiments are needed to establish a causal relation between the reduced immune effects of PTX3 and reduced tumor progression.
Author Response
The authors present a thorough mechanistic investigation on the pathophysiological role of PTX3 in the carcinogenesis, immunization and angiogenesis of fibrosarcoma. The research is mostly done in vivo and thoroughly designed and reported. As such, the study is certainly of interest for the readers of this journal. Some minor issues should be improved before publication is warranted:
The abstract has a minor typo: "The inflammation-associated long 27 pentraxin 3 (PTX3) was found to reduceS FGF-2-mediated angiogenesis"
DONE
Several sentences in the results section are borderline unreadable because the authors have chosen to include many means and standard deviations in the results text, e.g. "Indeed, the mean±SD of infiltrating Tryptase-positive mast cell morphometric values (Fig. 3) 137 were: 0.075±0.019 vs 0.027±0.009 in mock- vs PTX3-MC17-51 grafts (Fig. 3C) and 0.026±0.001 vs 138 0.003±0.001 for wild type MC17-51 lesions growing in WT vs TgN(Tie2-hPTX3) mice (Fig. 3F)." My suggestion is that it would be better to include these numbers as a small table in the figures itself so that the results text can be kept as clean as possible.
DONE, we have included the numbers in a small table in each figure.
In the introduction or results section, the authors should put more effort in describing the experimental relevance of the TgN(Tie2-hPTX3) mouse model. In the current version, this model is not explained and therefore readers may not fully understand the relevance of these experiments.
This part has been added in one new sentence in the Results and Discussion.
In the title of the last figure, the authors write "Figure 7. PTX3 overexpression reduces apoptosis.". However, they do not explain what the link between C3 and apoptosis would be. In addition, it is unclear to me how reduced apoptosis would lead to reduced tumor progression.
We apologize for the mistake, the title of this legend is wrong. The correct title has been updated. METTERE NELLA LEGENDA IL TITOLO NUOVO: “PTX3 overexpression reduces the deposition of the complement C3 component”.
The authors could more explicitly describe in the discussion that their results are largely correlative and what additonal experiments are needed to establish a causal relation between the reduced immune effects of PTX3 and reduced tumor progression.
This is already stated at the end of the Results, and a new part has been added to the Discussion.
Reviewer 2 Report
in the study ''PTX3 modulates neovascularization and immune inflammatory infiltrate in a murine model of fibrosarcoma'' Annese et al delinete the role of PTX3 on vascularization and development of fibrosarcoma using both PTX3 overexpressing xeonografts and a transgenic mouse model. the authors show that PTX3 expression is correlated with reduced tumor growth and tumor vascularization. Furthermore they show that PTX3 expression reduces the levels of immune cell migration to the tumors suggesting that tissue micro environment is affected by PTX3 levels. Since fibrosarcoma is rare the development of relevant disease models and targets is of interest to the general scientific community. I recommend publication in its current form.
Author Response
I thanks the REVIEWER for his/her positive comments.